# COVID-19 in lung transplant recipients—Risk prediction and outcomes

Jan C. Kamp[1,2]*, Jan B. Hinrichs[2,3], Jan Fuge[1,2], Raphael Ewen[1], Jens Gottlieb[1,2]

**1** Department of Respiratory Medicine, Hannover Medical School, Hannover, Germany, **2** Biomedical Research in Endstage and Obstructive Lung Disease Hannover (BREATH), German Center for Lung Research (DZL), Giessen, Germany, **3** Department of Diagnostic and Interventional Radiology, Hannover Medical School, Hannover, Germany

* Kamp.Jan-Christopher@mh-hannover.de

## Abstract

Patients after lung transplantation are at risk for life-threatening infections. Recently, several publications on COVID-19 outcomes in this patient population appeared, but knowledge on optimal treatment, mortality, outcomes, and appropriate risk predictors is limited. A retrospective analysis was performed in a German high-volume lung transplant center between 19th March 2020 and 18th May 2021. Impact of COVID-19 on physical and psychological health, clinical outcomes, and mortality were analyzed including follow-up visits up to 12 weeks after infection in survivors. Predictive parameters on survival were assessed using univariate and multivariate proportional hazards regression models. **Out of 1,046 patients in follow-up, 31 acquired COVID-19 during the pandemic. 12 of 31 (39%) died and 26 (84%) were hospitalized**. In survivors a significant decline in exercise capacity (p = 0.034), TLC (p = 0.02), and DLCO (p = 0.007) was observed at follow-up after 3 months. Anxiety, depression, and self-assessed quality of life remained stable. Charlson comorbidity index predicted mortality (HR 1.5, 1.1–2.2; p = 0.023). In recipients with pre-existing CLAD, mortality and clinical outcomes were inferior. However, pre-existing CLAD did not predict mortality. COVID-19 remains a life-threatening disease for lung transplant recipients, particularly in case comorbidities. Further studies on long term outcomes and impact on pre-existing CLAD are needed.

## 1. Introduction

Since the beginning of the ongoing worldwide severe acute respiratory syndrome coronavirus 2 (SARS-CoV-2) pandemic, more than 172 million confirmed cases and more than 3.7 million deaths were documented by the World Health Organization (WHO) in the report of June 9, 2021 [1].

The clinical presentation of people infected by SARS-CoV-2 is termed as Coronavirus disease 2019 (COVID-19). Most patients (approximately 80%) suffer from mild to moderate symptoms and recover after the initial disease phase, which is dominated by symptoms and signs of viral response like fever, dry cough, and lymphopenia. However, a considerable

**Data Availability Statement:** The data presented in this study are available only on reasonable request from the corresponding author or from the German Center for Lung Research (DZL) data warehouse.

[Internet: "https://dzl.de/en/dzl-data-warehouse/"]. Contacts for data management requests: [Internet: "https://dzl.de/en/platforms/contact-central-biobanking-management/"]. General public deposition of patient data is restricted according to the written informed consent which has been given from all patients for the use of their data for scientific purposes. Approval was given by the local institutional review board at Hannover Medical School (no. 2923-2015).

**Funding:** Dr. Kamp is supported by PRACTIS – Clinician Scientist Program of Hannover Medical School, funded by the German Research Foundation (DFG, ME 3696/3-1). www.dfg.de The funders had no role in study design, data collection and analysis, decision to publish, or preparation of the manuscript.

**Competing interests:** The authors have declared that no competing interests exist.

**Abbreviations:** CCI, Charlson Comorbidity Index; CLAD, chronic lung allograft dysfunction; COVID-19, Coronavirus disease 2019; DLCO, diffusing capacity for carbon monoxide; EQ-5D-5L, Euro Quality of Life 5 dimensions 5 levels questionnaire; $FEV_1$, forced expiratory volume in 1 second; GAD, generalized anxiety disorder; HR, hazard ratio; IQR, interquartile range; LTx, lung transplantation; MDD, major depressive disorder; PHQ-4, patient health questionnaire 4; QoL, quality of life; SARS-CoV2, severe acute respiratory syndrome coronavirus 2; TLC, total lung capacity; WHO, World Health Organization; WHO-OS, World Health Organization Ordinal Scale.

amount of patients develop increasing host inflammatory response symptoms after the initial phase with dyspnea and hypoxemia up to life-threatening organ impairment like acute respiratory distress syndrome, shock, and cardiac failure [2]. Prolonged disease courses and persistent physical and psychological impairment after COVID-19 are well described in the general population [3] as well as in distinct pulmonary diseases [4].

COVID-19 in solid organ or lung transplant recipients was reported in cohort studies and case series [5–12] with a case fatality rate of 10–46%. There is a lack of knowledge about the preferred treatment strategy as well as long term physical and psychological outcomes in this patient population. Moreover little is known about the impact of COVID-19 on pre-existing chronic lung allograft dysfunction (CLAD).

Aim of this study was to contribute to the still limited knowledge on clinical outcomes in lung transplant (LTx) recipients with a confirmed infection by SARS-CoV-2. Moreover, potential predictors of mortality and psychological outcomes were analyzed as reliable data on these issues are still lacking.

## 2. Methods

### 2.1 Setting

A retrospective analysis was performed in a single high-volume lung transplant center. The LTx outpatients' clinic actively follows 1,046 patients after unilateral, bilateral or combined lung transplantation by regular outpatient and telemedical visits. All patients received center-based blood samples (for cytomegalovirus and immunosuppressive drug monitoring) and telephonic advice to transplant-related issues as case management. Basically, all of our patients are encouraged to inform our outpatients' clinic on occurring relevant health-related events like respiratory infections.

All patients fulfilled the following inclusion criteria: i) follow-up since lung transplantation in our outpatients' clinic, ii) SARS-CoV-2 infection, confirmed by polymerase chain reaction-based assay until 13th March 2021, and iii) follow-up in our outpatients' clinic 4–12 weeks after survived SARS-CoV-2 infection or death (i.e. after hospital discharge in inpatients / after end of symptoms in outpatients).

All SARS-CoV-2 infected outpatients were regularly followed after diagnosis by daily video consultations (Medflex GmbH, Konstanz, Germany) [13]. All LTx patients with COVID-19 hospitalized in other hospitals were managed by daily telephonic consultation with physicians on charge in these hospitals. Baseline parameters were derived by last available visit prior to infection. Comorbidity burden was assessed by the Charlson Comorbidity Index (CCI) [14]. This index includes several comorbid conditions which might alter the risk of mortality. A point value between 0 and 29 is calculated with a higher value indicating higher level of comorbidity.

Disease severity of SARS-CoV2 infection was assessed by the World Health Organization Ordinal Scale (WHO-OS) [15]. Non-hospitalized patients without / with limitation on activities were classified as "1" / "2", hospitalized patients without / with conventional oxygen therapy requirement as "3" / "4", patients with need of non-invasive ventilation or high flow oxygen therapy as "5", patients with need of intubation and mechanical ventilation as "6", patients in whom renal replacement therapy, administration of vasoactive drugs or extracorporeal membrane oxygenation therapy was required as "7", and patients who died from acute SARS-CoV2 infection as "8". In all patients who survived, follow-up parameters were assessed 4 to 12 weeks after infection. Observation period ended on 18th May 2021.

All patients provided written informed consent for the use of their data for scientific purposes. Approval was given by the local institutional review board (no. 2923–2015).

On on-site visits routine questionnaires, spirometry, chest X-ray, and routine laboratory testing incl. calcineurin inhibitor trough levels and cytomegalovirus-testing were performed. Pulmonary function testing and measuring of diffusing capacity for carbon monoxide (DLCO) was conducted according to current European Respiratory Society / American Thoracic Society standards [16, 17].

CLAD diagnosis and staging were conducted according to current International Society of Heart and Lung Transplantation recommendations [18].

During regular on-site follow-up all patients completed a self-reported questionnaire including need of medical aids (such as oxygen therapy, non-invasive ventilation, rollator, wheel chair, and feeding tube) and exercise capacity which is determined based on no. of flight of stairs.

Mental Health and quality of life (QoL) were assessed using the Euro Quality of Life 5 dimensions 5 levels questionnaire (EQ-5D-5L) [19] and the patient health questionnaire 4 (PHQ-4) [20]. The EQ-5D-5L has five sub scores in 5 dimensions ranging from 1 to 5 points and a QoL visual analogue scale (QoL-VAS) with a range from 0 (worst health one can imagine) to 10 (best health one can imagine). The PHQ-4 has 4 questions in two dimensions (anxiety and depression) ranging from 0 to 3 points, respectively. PHQ-4 scores are summed up in total and separately for generalized anxiety disorder (GAD, GAD-2) and major depressive disorder (MDD, PHQ-2). A GAD-2 or PHQ-2 score greater than 3 indicates signs of GAD or MDD, respectively.

## 2.2 Statistical analysis

We used IBM SPSS Statistics 27.0 (IBM Corp, Armonk, NY, USA) and Stata 13.0 (State Corp LP, College Station, Texas, USA) statistical software to analyze the data. Most continuous variables are shown as median and 25%–75% interquartile range (IQR). Only parameters from the EQ-5D-5L and PHQ-4 are shown as mean and standard deviation in case of small case numbers. Categorical variables are shown as numbers and percent (%). For comparisons of patient time points, Fisher's exact test, Chi-square test, paired Wilcoxon-test, McNemar Bowkers test or paired t-tests were used as appropriate. Kaplan-Meier survival estimates were created for the whole group. To determine hazards of death proportional hazards regression models were created (one univariate and one multivariate model). All reported p-values are two sided and derived from paired data. P-values <0.05 were considered statistically significant.

## 3. Results

A total of 31 LTx recipients with confirmed infection by SARS-CoV-2 were included in this analysis (Fig 1). Patient characteristics at the onset of symptoms are shown in Table 1. Fifty-eight percent of the patients were female and median age was 54 (47.5/58) years. CCI was 4 (4/5.5) indicating a substantial burden of comorbid conditions and n = 14 (45%) patients suffered from preexisting CLAD. Until the end of the observation period additional 7 patients acquired SARS-CoV-2 infection but with insufficient follow-up (1 died).

Median WHO-OS classification during the acute disease period was 4 (4/7) indicating predominantly severe disease courses and mortality was high (n = 12, 39%, Fig 2). Causes of death were respiratory failure from COVID-19 or from COVID-19 and progressive CLAD (n = 5, 42% and n = 3, 25%, respectively) as well as fatal complications following COVID-19 (necrotizing pancreatitis; n = 1; acute liver failure, n = 1; and septic shock, n = 2). Twenty-six (84%) patients were hospitalized due to dyspnea, n = 16 (61%); decrease in oxygen saturation, n = 6 (23%); diarrhoea, n = 1 (4%); or other reasons, n = 3 (12%). Median time from onset of symptoms to hospitalization was 7 (3/9) days and median duration of hospitalization was 19 (11/29)

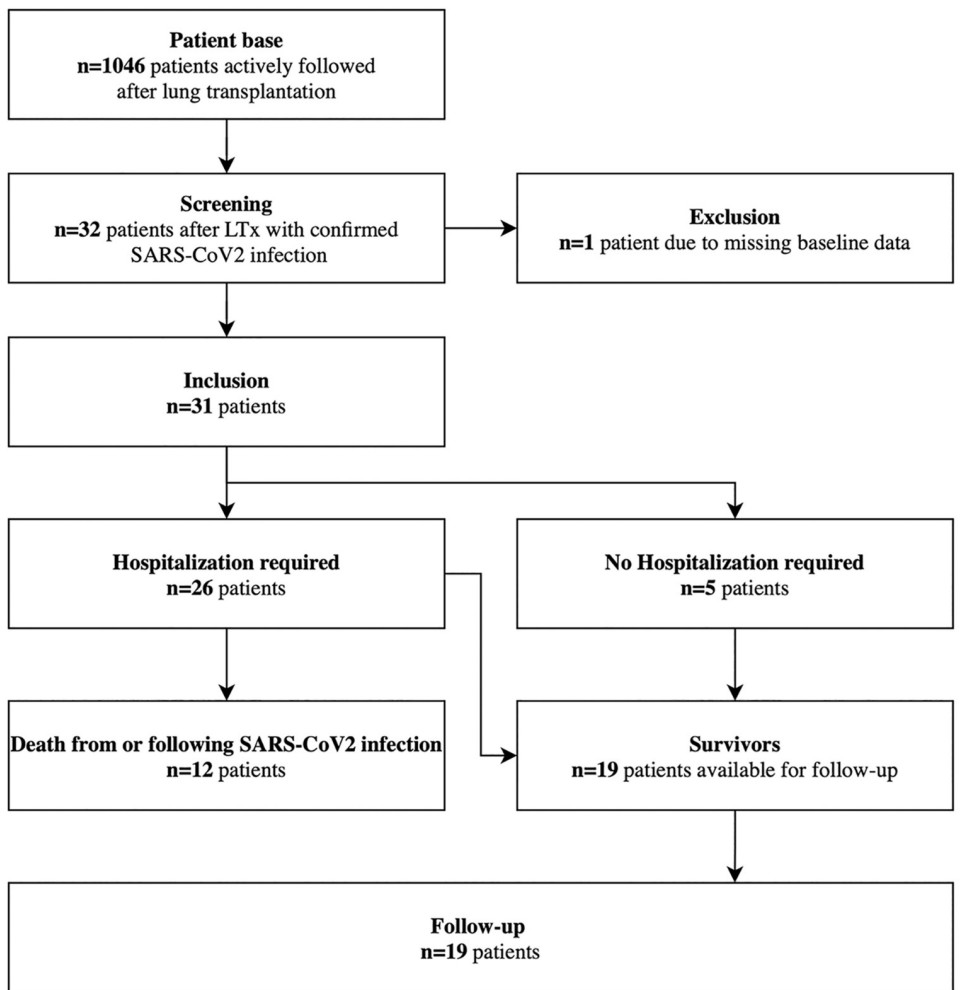

**Fig 1. Flow chart.** SARS-CoV2, severe acute respiratory syndrome coronavirus 2.

days. Median time from onset of symptoms to oxygen therapy / death was 8 (3/9) and 27 (12/48) days, respectively.

Key characteristics of COVID-19 courses are presented in Table 2. In 18 (58%) patients 70 video consultations (median 3 per patient, 2/7.5) were performed since the onset of SARS-CoV-2 related symptoms. Of these patients, 2 were hospitalized immediately after the first video consultation and 11 following a later visit. Fifteen (58%) patients needed intensive care. Conventional oxygen therapy was required in all hospitalized patients (n = 26, 84%), high flow oxygen therapy in n = 3 (10%) patients, non-invasive ventilation in n = 5 (16%) patients, mechanical ventilation in 8 (26%) patients, and extracorporeal membrane oxygenation therapy in n = 4 (13%) patients. Twenty-two (71%) patients received dexamethasone, n = 5 (16%) patients received remdesivir, and n = 1 (3%) patient was treated with reconvalescent plasma. Cell-cycle inhibitors were withheld during hospitalization in all patients until discharge and thromboprophylaxis / anticoagulants were administered according to the treating physicians' appraisal.

Nineteen patients underwent 70 (median 3, 2/7) video consultations after infection. In n = 6 patients decrease in oxygen saturation to <92% without signs of dyspnea (so called

**Table 1. Patient characteristics at the onset of symptoms (31 patients).**

| | |
|---|---|
| Gender; n (%) | |
| • Female | • 18 (58) |
| Age at diagnosis of SARS-CoV-2 infection | 54 (47.5/58) |
| BMI at diagnosis of SARS-CoV-2 infection | 22.3 (19.5/23.95) |
| Charlson Comorbidity Index at diagnosis of SARS-CoV-2 infection | 4 (4/5.5) |
| Diagnosis leading to transplantation; n (%) | |
| • Interstitial lung disease | • 14 (45) |
| • Emphysema | • 9 (29) |
| • Cystic fibrosis / Primary ciliary dyskinesia | • 4 (13) |
| • Pulmonary hypertension | • 3 (10) |
| • Other | • 1 (3) |
| Lung transplantation; n (%) | |
| • Unilateral | • 2 (6) |
| • Bilateral | • 28 (90) |
| • Combined heart and lung transplantation | • 1 (3) |
| Duration from lung transplantation to onset of symptoms (years) | 7.9 (3.2/12.9) |
| Primary immunosuppressive therapy; n (%) | |
| • Tacrolimus | • 24 (77) |
| • Ciclosporine | • 7 (23) |
| Other immunosuppressive therapy; n (%) | |
| • Mycophenolate-mofetil | • 25 (81) |
| • Everolimus | • 4 (13) |
| • Azathioprine | • 2 (6) |
| • Prednisolone | • 31 (100) |
| Preexisting CLAD; n (%) | 14 (45) |
| • Stage 1 / 2 | • 10 (32) |
| • Stage 3 / 4 | • 4 (13) |
| CLAD therapy; n (% of CLAD patients) | |
| • Azithromycin | • 14 (100) |
| • Montelukast | • 6 (43) |
| • Extracorporeal photopheresis | • 9 (64) |

Continuous variables are presented as median and 25% / 75%r quartiles; SARS-CoV-2, severe acute respiratory syndrome corona virus-2; BMI, body mass index; CLAD, chronic lung allograft dysfunction.

"silent hypoxemia") was detected 10 (10/12) days after onset of symptoms leading to hospital admission. In these 6 patients, duration of hospital stay was 17 (12/24) days and n = 4 (66%) patients needed intensive care treatment. In those patients who underwent more than 2 video consultations no trends in vital signs were detectable.

Follow-up results are displayed in Table 3. Forced expiratory volume in 1 second ($FEV_1$) decline >10% was seen in 3 (16%) patients at follow-up. Total lung capacity (TLC) declined significantly (98, 94/102 vs. 86, 80/100; p = 0.02) as well as mean DLCO (66, 52.5/84 vs. 49, 44/59; p = 0.007). Exercise capacity, measured by no. of flight of stairs, worsened significantly (p = 0.034).

Regarding the subgroup of LTx recipients with pre-existing CLAD (n = 14) there was a substantial deterioration in graft function. N = 6 (43%) of 14 patients died from or following COVID-19 and further n = 6 (43%) patients showed a decline ≥10% in at least one of FEV1,

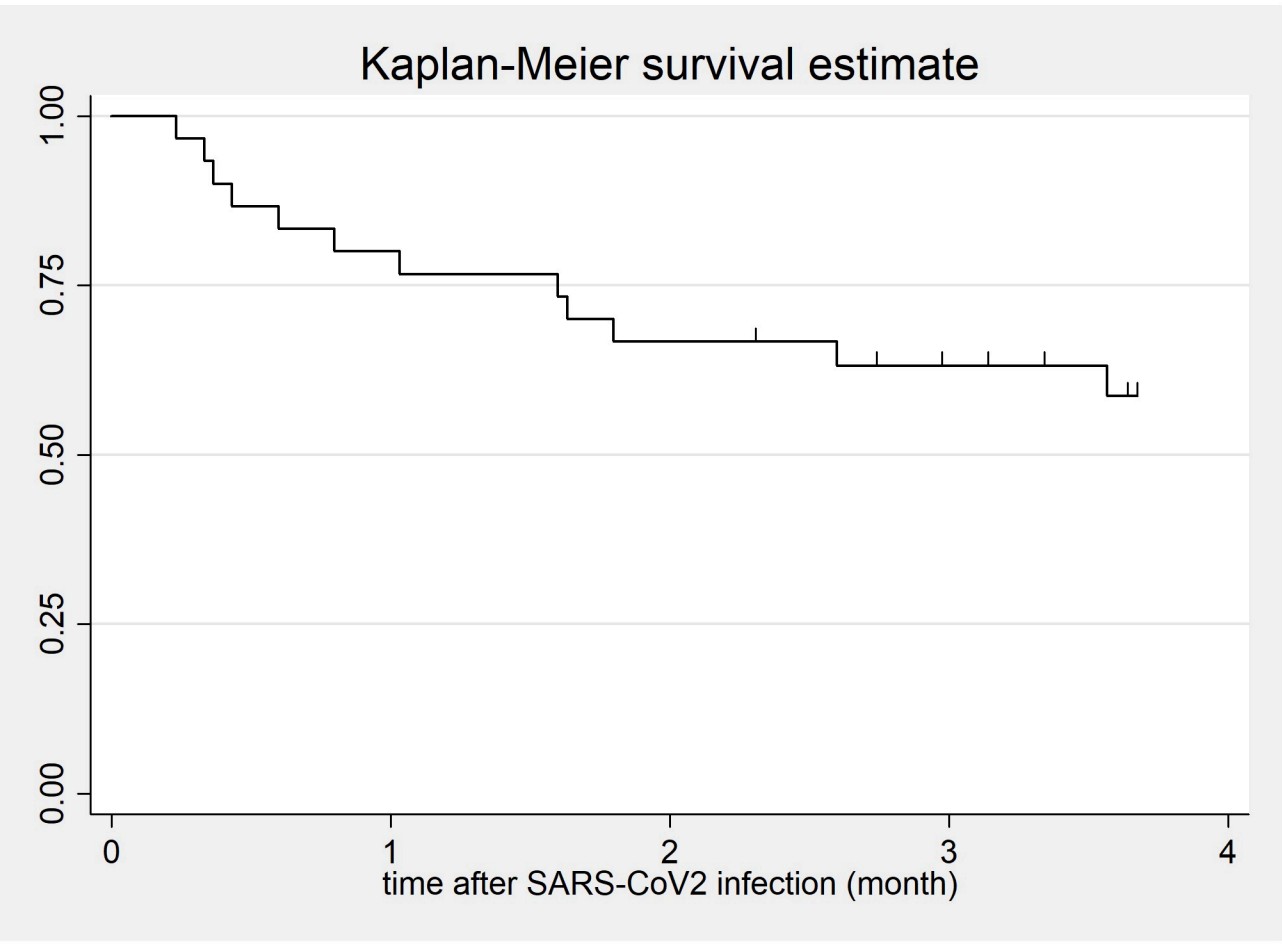

**Fig 2. Survival after COVID-19.** Kaplan-Meier survival estimate on 31 lung transplant recipients with confirmed SARS-CoV-2 infection.

TLC, and DLCO at follow-up signaling CLAD progression. In survivors without pre-existing CLAD no new CLAD diagnoses were made at follow-up.

Quality of life as measured by EQ-5D-5L absolute value tendentially worsened (88 ±14.3 vs. 80.8 ±21.4, p = 0.055) which was mostly reflected by the subscores for mobility (p = 0.066) and anxiety / depression (p = 0.033). The PHQ-4 as well as its subscores (PHQ-2 and GAD-2) in absolute and categorical values did not change significantly. Self-reported QoL-VAS results remained unchanged.

Results of the univariate and multivariate analyses are shown in Table 4. Univariate analysis demonstrated no elevated hazard on sex, body mass index, chronic kidney disease, coronary heart disease, dyslipidemia, arterial hypertension, diabetes mellitus, pre-existing CLAD, established azithromycin therapy, % FEV$_1$ predicted or % TLC predicted, and exercise capacity for mortality. Possible impact on mortality was found for age (Hazard Ratio, HR 1.1 (1.0–1.1); p = 0.029) and CCI (HR 1.5 (1.1–2.2); p = 0.023). A trend was visible for hemodialysis (HR 3.5 (0.9–13.1); p = 0.065) which was present in just 4 cases. Multivariate proportional hazards analyses testing for covariates age, CCI, and hemodialysis revealed no impact on mortality hazard for age and hemodialysis while CCI predicted mortality (HR 1.5 (1.1–2.2); p = 0.023).

**Table 2. Key characteristics of COVID-19 courses.**

| | |
|---|---|
| Hospitalization | 26 (87) |
| • ICU treatment | • 15 (48) |
| Time from onset of symptoms to hospitalization (d), n = 26 | 7 (3/9) |
| Reasons for admission, n = 26 | |
| • Dyspnea | • 16 (61) |
| • Decrease in oxygen saturation, measured during video consultation | • 6 (23) |
| • COVID-19-associated diarrhea | • 1 (4) |
| • Other conditions, not associated with COVID-19 | • 3 (12) |
| Duration of hospitalization (days) | 19 (11/29) |
| Mortality | 12 (39) |
| Cause of death, n = 12; n (% of deceased patients) | |
| • Respiratory failure from COVID-19 and (if applicable) progressive CLAD | • 8 (67) |
| • Fatal complications following COVID-19 | • 4 (34) |
| Time from onset of symptoms to death (days), n = 12 | 27.5 (12.5/48.25) |
| Time from onset of symptoms to oxygen therapy (days), n = 26 | 8 (3/9) |
| CLAD Progression; n (% of 14 CLAD patients) | |
| • Decline ≥10% in at least one of $FEV_1$, TLC, and DLCO | • 6 (43) |
| • Death from or following COVID-19 | • 6 (43) |

Categorical variables are presented as n and percent unless stated otherwise; ICU, intensive care unit; COVID-19, Corona virus disease 19; $FEV_1$, forced exhaled volume in 1 second; TLC, total lung capacity; DLCO, diffusing capacity for carbon monoxide.

## 4. Discussion

We here report the clinical courses of a group of 31 LTx recipients from our institution diagnosed with SARS-CoV-2 infection. We determined the CCI independent risk factors for poor outcomes in LTx recipients with SARS-CoV-2 infection. Hospitalization rate was high and mostly due to COVID-19-associated dyspnea. More than half of all hospitalized patients needed intensive care indicating a high average disease severity. More than one third of our patients died.

Mortality rates from most recently published works were comparable. Saéz-Giménez et al. presented data on short-term outcomes of 44 LTx recipients with confirmed COVID-19 and hospitalization to one of six Spanish referral centers with a mortality rate of 39% [7]. Kates et al. found a mortality rate of 33% in 30 LTx recipients from the United States [9]. Coll et al. reported a mortality rate of 46% in 50 LTx recipients with COVID-19 from Spain [10]. Aversa et al. reported on 32 LTx recipients with COVID-19 from New York City with a mortality rate of 34% [11]. Further small case series on the outcomes of lung transplant recipients after COVID-19 revealed mortality rates between 10% and 25% [6, 8].

Long-term outcomes in LTx recipients with COVID-19 were published recently in a French cohort. Substantially lower mortality (14.3%) was found in 35 LTx recipients [5]. The difference in mortality between the French cohort and our patients might be explained by the sum of the following points: i) the minor amount of CLAD patients compared to our cohort (20% vs. 45%); ii) lower median values in age and body mass index (48.1 and 21.5 vs. 54 and 22.3, respectively); iii) the earlier observation period (March, 1 2020 to May, 19, 2020 compared with March 19, 2020 and May, 18, 2021). Most fatal cases from our cohort were diagnosed

**Table 3. Clinical outcomes in survivors, obtained 4–12 weeks after SARS-CoV-2 infection.**

| | Baseline* | Follow-Up** | p-value |
|---|---|---|---|
| Time from baseline to symptom onset / Time from symptom onset to follow-up, n = 19 (days) | 84 (13–175) | 87 (69–143) | n/a |
| $FEV_1$, n = 19 (% baseline) | 92 (80.5/96) | 93 (74.5/100) | 0.247 |
| TLC, n = 15 (% baseline) | 98 (94/102) | 86 (80/100) | **0.020** |
| DLCO, n = 11 (% baseline) | 66 (52.5/84) | 49 (44/59) | **0.007** |
| QoL-VAS, n = 19 (points of 10) | 7 (6/8.5) | 8 (6.5/8) | 0.938 |
| Exercise capacity (no. of flight of stairs), n = 19, n (%) | | | **0.034** |
| • >2 | • 8 (42) | • 5 (26) | |
| • 2 | • 6 (32) | • 6 (32) | |
| • 1 | • 5 (26) | • 5 (26) | |
| • 0 | • 0 (0) | • 3 (16) | |
| Need of medical aids at follow-up, n = 19, n (%) | | | n/a |
| • Oxygen therapy | • 2 (11) | • 3 (16) | |
| • NIV therapy | • 0 (0) | • 0 (0) | |
| • Rollator | • 5 (26) | • 1 (5) | |
| • Wheel chair | • 0 (0) | • 4 (21) | |
| • Feeding tube | • 1 (5) | • 1 (5) | |
| EQ-5D-5L, n = 19 (%) | 88.0 ±14.3 | 80.8 ±21.4 | 0.055 |
| • Mobility (points) | • 1.4 ±0.8 | • 2.0 ±1.1 | 0.066 |
| • Self-care (points) | • 1.3 ±0.5 | • 1.5 ±0.8 | 0.096 |
| • Usual activities (points) | • 1.6 ±0.8 | • 1.8 ±1.2 | 0.236 |
| • Pain / Discomfort (points) | • 1.5 ±0.7 | • 1.8 ±1.2 | 0.163 |
| • Anxiety / Depression (points) | • 1.5 ±0.8 | • 1.9 ±1.0 | **0.033** |
| PHQ-4, n = 19 (points) | 2.4 ±2.5 | 2.7 ±2.8 | 0.261 |
| • PHQ-2 (points) | • 1.0 ±1.1 | • 1.6 ±1.5 | 0.090 |
| • GAD-2 (points) | • 1.4 ±1.5 | • 1.2 ±1.5 | 0.305 |
| PHQ-2—Depressive Disorder, n (%) | 2 (11) | 5 (26) | 0.250 |
| GAD-2—Anxiety Disorder, n (%) | 3 (16) | 3 (16) | 1.000 |

Continuous variables are presented as median and 25% / 75% inter quartile range (IQR) or mean and standard deviation as appropriate; categorical variables are presented as n and percent unless stated otherwise;

*Baseline was defined as last available visit prior to SARS-CoV-2 infection;

**Follow-up visits were conducted 4–12 weeks after survived SARS-CoV-2 infection;

COVID-19, Corona virus disease 19; ICU, intensive care unit; $FEV_1$, forced exhaled volume in 1 second; TLC, total lung capacity; DLCO, diffusing capacity for carbon monoxide; QoL-VAS, quality of life, assessed by visual analogue scale; EQ-5D-5L, Euro-QoL 5 dimensions 5 levels questionnaire; PHQ-4, patient health questionnaire 4; PHQ-2, patient health questionnaire 2; GAD-2, generalized anxiety disorder questionnaire 2; CLAD, chronic lung allograft dysfunction.

between November 2020 and February 2021 so a relation to the British / B.1.1.7 / alpha variant is not excludable what might contribute to the differing mortality rates [21].

Recently, a large registry study from the Unites States by Hady et al. was published [12]. Therein, 2,307 solid organ transplant recipients were propensity matched for demographics and comorbidities to a huge cohort from the general population. Interestingly, no significant differences regarding outcomes were found. However, LTx recipients were underrepresented in this cohort (4.7% of patients) and no survival estimate for this subgroup was stated. The authors suggest that increased risk for mortality in solid organ transplant recipients might be primarily explained by the higher burden of comorbidities. We agree that high morbidity should be an independent risk factor but nevertheless the required highly potent

**Table 4. Results of univariate and multivariate linear regression analyses on predictive outcome parameters.**

| Item | Univariate | Multivariate |
|---|---|---|
| | HR (95% CI); p-value | HR (95% CI); p-value |
| Male Sex | 0.7 (0.2–2.4); p = 0.590 | |
| Age | 1.1 (1.0–1.1); **p = 0.029** | |
| Body mass index | 0.9 (0.8–1.1); p = 0.345 | |
| Chronic kidney disease | 2.6 (0.3–20); p = 0.356 | |
| Hemodialysis | 3.5 (0.9–13.1); p = 0.065 | |
| Coronary heart disease | 1.5 (0.3–6.8); p = 0.613 | |
| Dyslipidemia | 1.0 (0.3–3.2); p = 0.979 | |
| Arterial hypertension | 2.8 (0.6–12.9); p = 0.182 | |
| Diabetes mellitus | 1.2 (0.4–3.9); p = 0.802 | |
| CLAD | 1.2 (0.4–3.8); p = 0.732 | |
| Charlson comorbidity index | 1.5 (1.1–2.2); **p = 0.023** | 1.5 (1.1–2.2); **p = 0.023** |
| % FEV$_1$ predicted | 1.0 (1.0–1.0); p = 0.138 | |
| % TLC predicted | 1.0 (0.9–1.0); p = 0.152 | |
| Azithromycin therapy | 1.1 (0.4–3.6); p = 0.842 | |
| Exercise capacity* | 0.6 (0.3–1.2); p = 0.126 | |

HR, Hazard Ratio; CI, Confidence interval; CLAD, chronic lung allograft dysfunction; FEV$_1$, forced exhaled volume in one second; TLC, total lung capacity;

*Exercise capacity was assessed based on no. of flight of stairs.

immunosuppressive therapy in lung transplant recipients as well as the limited respiratory reserve and the substantial CLAD prevalence even contribute to poor outcomes in these vulnerable patients. In our cohort, high comorbidity rate is reflected by the CCI.

One key experience from daily conducted video consultations was that many patients were in seemingly stable clinical condition after the onset of SARS-CoV-2 symptoms for up to 10 days. Following this, we observed a sudden respiratory deterioration or even decrease in oxygen saturation without shortness of breath rapidly leading to intensive care requirement in many cases. Asymptomatic decrease in oxygen saturation is a well described feature of COVID-19 termed as silent hypoxemia [22].

Pathomechanisms described recently might apply to lung transplant recipients as well [2]. Pulmonary embolism is a common feature of COVID-19 but in our cohort we did not observe any thromboembolic events. This might be explained by the late peak of cases when risk of thromboembolic events in COVID-19 was already well known and most patients received anticoagulants.

After hospital admission, we often observed prolonged lengths of stay (median 19, 11/29 days), especially in those patients who finally died (median time from onset of symptoms to death 27, 12/48 days).

These data considerably differ from those reported by Karagiannidis et al. in a large study on in-hospital mortality of patients from the German general population with COVID-19 [23]. In this work, in-hospital mortality was 22% (compared to 46% in our study) and hospital stays were significantly shorter (14 vs. 19 days) what might potentially be explained due to the therapeutics engaged in LTx recipients (e.g. exctracorporeal therapies) rather than in the general population. Moreover, half of all deaths occurred within 12 days after admission what is inconsistent with our data as described above.

Recently, Permpalung et al. reported on clinical outcomes after COVID-19 in a 1:2 matched case-control study with 24 SARS-CoV-2 infected and 48 non-infected LTx recipients. In that

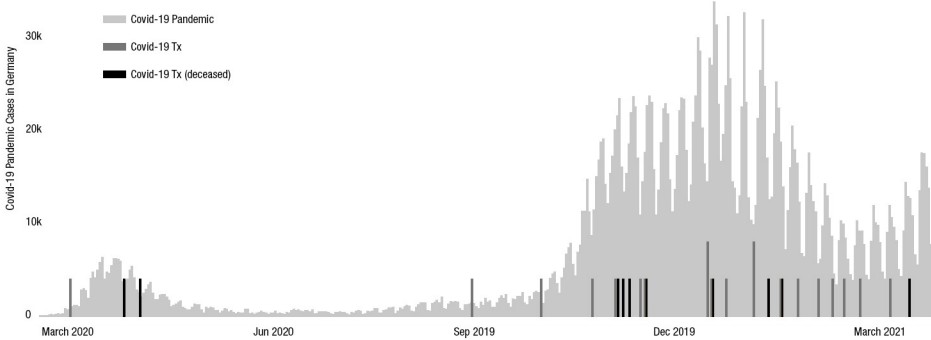

**Fig 3. Incidence of SARS-CoV-2 infections in Germany.** COVID-19 Pandemic subsumes the incidence in the German general population; COVID-19 Tx subsumes all confirmed SARS-CoV-2 infections in lung transplant recipients from our institution; those who deceased are shown black.

work, elevated risk for secondary infections and re-hospitalization was found within 90 days post COVID-19. However, no impact on CLAD or on the rate of acute cellular or antibody mediated rejections and was detected [24].

The incidence of SARS-CoV-2 infections in our patient-cohort during the observation period was similar to nationwide German incidence (Fig 3). However, it might be possible that the number of unreported / undetected cases were considerably higher in the general population. Considering the incidence and mortality in our LTx recipients over time a noticeable peak occurred between November 2020 and February 2021. Most confirmed cases and fatal outcomes arose from this time frame. As far as we know, no comprehensive sequencing of SARS-CoV-2 variants has been performed in our patients but it can be speculated that this cluster might be associated with the alpha variant B.1.1.7 which gained increasing national relevance during this period [22].

Follow-up diagnostics in survivors performed 4–12 weeks after terminated infection revealed significant lung function impairment and reduced exercise capacity. The relatively stable findings in $FEV_1$ after COVID-19 are standing in contrast to infections caused by common community-acquired respiratory viruses which are known to cause a mean decline in $FEV_1$ of approximately 15% after 28 to 90 days [25]. However, TLC and DLCO decreased significantly indicating a restrictive airway disease and interstitial alterations due to inflammatory and post-inflammatory processes.

The most important limiting single factor of long-term survival in LTx recipients is CLAD, responsible for >40% of deaths beyond the first year (17). It might be expected that LTx recipients with pre-existing CLAD are at particular high risk of poor outcome when suffering from COVID-19 as with other chronic respiratory disease (24). Almost half of CLAD patients died from or following COVID-19 and the same amount suffered from progressive deterioration in graft function at follow-up. Only 2 recipients (14%) with pre-existing CLAD showed stable findings at follow-up indicating a potential extremely vulnerable subgroup. In some patients decline in lung function was reflected by persistent alterations in chest computed tomography. As previously described by Messika et al. [5] radiologic imaging findings due to COVID-19 are comparable between LTx recipients and non-immunocompromised patients. However, it should be mentioned that there is a diagnostic uncertainty regarding the differentiation between progression of a preexisting CLAD with restrictive allograft syndrome or mixed phenotype according to current recommendations [18] and a (post-) COVID-19 pattern. Images

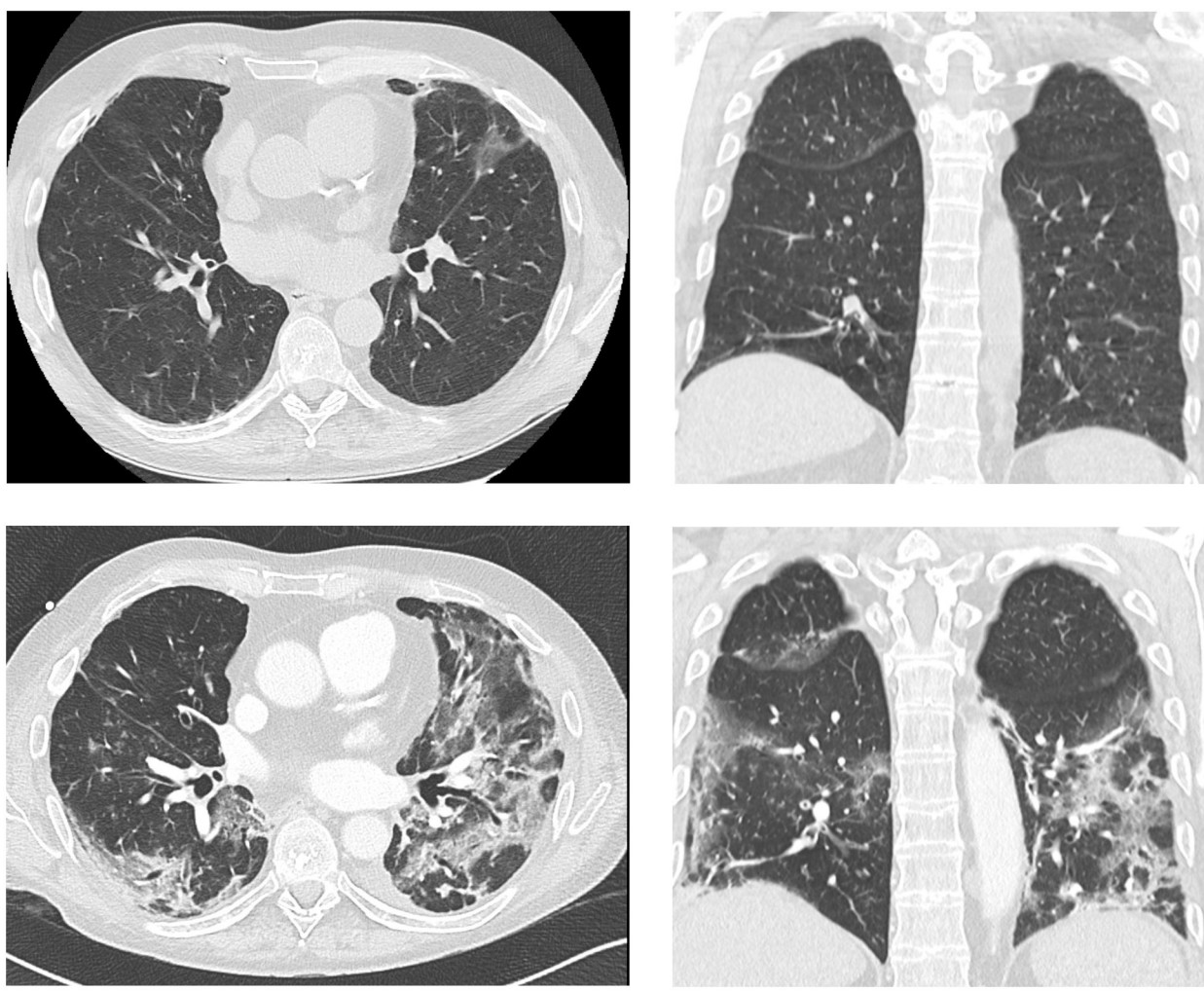

**Fig 4. Chest computed tomography findings pre and post COVID-19.** Presentation of a 54-year-old male lung transplant recipient with pre-existing mixed phenotype chronic lung allograft dysfunction. A, B: High-resolution computed tomography (HR-CT) 4 months prior to COVID-19 infection. Axial slice showing distinct pleuroparenchymal infiltrates on the left side (arrow) and coronal slice showing streak densities in the basal parts of the lung as mild signs of CLAD. C, D: Axial HR-CT obtained 2 weeks after survived infection showing areas auf consolidation (arrows) mixed with ground glass opacities (arrow head). Images prior to COVID-19 were generated on a multidetector CT according to a standard protocol using a 64-row scanner (GE Lightspeed VCT, GE Healthcare, Chalfont St. Giles, United Kingdom). Either 100 kVp or 120 kVp tube voltage was used adjusted to the body mass index of the patient. Briefly, we used a detector collimation of 64 0.625 mm and a reconstructed slice thickness of 1.25 mm. High resolution CT images following COVID-19 were generated on a dual source 2x96-row MDCT (Siemens FORCE, Siemens Healthineers, Forchheim, Germany) (detector collimation: 192x0.7 mm, slice thickness: 1 mm, interval: 0.7 mm) and reconstructed in an appropriate kernel. COVID-19, Coronavirus disease 2019; CLAD, chronic lung allograft dysfunction; CT, computed tomography.

from chest computed tomography in one of our LTx recipients with pre-existing CLAD before and after COVID-19 are presented in Fig 4.

Today, there is still a lack on reliable mortality predictors for LTx recipients with imminent or confirmed SARS-CoV-2 infection. Messika et al. recently described a relationship between overweight and risk of death in this patient population [5]. Our analyses revealed the CCI as a marker of morbidity and the WHO-OS as a severity marker during COVID-19 as predictive markers on mortality before and during COVID-19, respectively. Certainly, the latter finding is not surprising but at least confirming the prognostic value of the WHO-OS in LTx recipients suffering from COVID-19. However, the CCI as an appropriate prognostic tool for this patient

population is a very relevant finding and it would be worth to evaluate it in larger studies. Actually, these findings support the results of Hady et al. regarding the strong association between comorbidities and mortality in solid organ recipients [12].

None of our patients was vaccinated prior to infection. However, until the end of the observation period further LTx recipients were diagnosed with SARS-CoV-2 infections three of whom received their first vaccination, one with VaxZevria® (Astra Zeneca) and two with Comirnaty® (Biontech/Pfizer). There is concern about limited immune response to SARS--CoV-2 vaccines in transplant recipients [26]. Further studies are needed to evaluate efficacy of these vaccines and preferable dose regimens in these patients.

There are some limitations of our study. First the short observation period after terminated SARS-CoV-2 infection, second the small number of patients, third the monocentric design, and fourth the lack of a control group. Future studies are needed to analyze long-term survival after COVID-19 in LTx recipients, the influence on pre-existing CLAD, as well as the response to SARS-CoV-2 vaccinations in this patient population.

To date, many issues on COVID-19 in lung transplant recipients remain unanswered, especially the relevance of new virus variants and the benefit of future drug developments. More studies are needed to optimize treatment and vaccination strategies.

In conclusion, mortality and morbidity of COVID-19 in lung transplant recipients is high, especially in case of a pre-existing CLAD and significant co-morbidities. Future studies are needed to assess long-term influence on physical and psychological health and especially on the risk of CLAD.

## Acknowledgments

We thank our coordinators from the transplant outpatients' department, Linda Häsler, Anita Fuhrmann, and Bianca Rink for the helpful support during this project.

## Author Contributions

**Conceptualization:** Jan C. Kamp, Jens Gottlieb.

**Data curation:** Jan Fuge.

**Formal analysis:** Jan C. Kamp, Jan Fuge.

**Investigation:** Jan C. Kamp, Jan B. Hinrichs, Jens Gottlieb.

**Methodology:** Jan C. Kamp, Jan Fuge, Jens Gottlieb.

**Project administration:** Jan C. Kamp.

**Visualization:** Jan B. Hinrichs.

**Writing – original draft:** Jan C. Kamp.

**Writing – review & editing:** Jan B. Hinrichs, Jan Fuge, Raphael Ewen, Jens Gottlieb.

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
