## [Decision Letter · Decision Letter 0]

9 Aug 2021

PONE-D-21-23993

COVID-19 in lung transplant recipients - risk prediction and outcomes

PLOS ONE

Dear Dr. Kamp,

Thank you for submitting your manuscript to PLOS ONE. After careful consideration, we feel that it has merit but does not fully meet PLOS ONE’s publication criteria as it currently stands. Therefore, we invite you to submit a revised version of the manuscript that addresses the points raised during the review process.

Please revise accordingly.

We look forward to receiving your revised manuscript.

Kind regards,

Academic Editor

PLOS ONE

Journal Requirements:

"Dr. Kamp is supported by PRACTIS – Clinician Scientist Program of Hannover Medical School, funded by the German Research Foundation (DFG, ME 3696/3-1). www.dfg.de The funders had no role in study design, data collection and analysis, decision to publish, or preparation of the manuscript"

Reviewers' comments:

Reviewer's Responses to Questions

**Comments to the Author**

1. Is the manuscript technically sound, and do the data support the conclusions?

Reviewer #1: Yes

Reviewer #2: Yes

2. Has the statistical analysis been performed appropriately and rigorously? 

Reviewer #1: Yes

Reviewer #2: Yes

3. Have the authors made all data underlying the findings in their manuscript fully available?

Reviewer #1: Yes

Reviewer #2: Yes

4. Is the manuscript presented in an intelligible fashion and written in standard English?

Reviewer #1: Yes

Reviewer #2: Yes

5. Review Comments to the Author

Reviewer #1: The subject included in the manuscript is of the utmost importance for broad medical community. Firstly, COVID-19 is the novel disease of unknown long-term consequences. Secondly, undoubtedly it will be present in population for many years, if not decades.

Therefore every report sheds a new light on the condition we will be facing in the years to come.

It is reviewer satisfaction to be able to congratulate the Authors. The forwarded paper is meticulous, comprehensive, and precise. It is astounding that out of more than 1000 LTx individuals, solely more then 30 have gotten ill.

Definitely this indicates accurate measures introduced to prevent them from being infected, some of them you have mentioned in the manuscript.

I am quite aware of the leading role of the Hannover Medical School in Europe in lung transplantation. Therefore I got impression the COVID-19 is not affecting seriously this population. It is reassuring message, which perhaps might be more openly emphasized.

In the discussion section you have made an extensive comparison with other LTx centers.

Generally speaking reading this manuscript has been a teachable moment.

Reviewer #2: Thank you for submititng this retrospective analysis of one of the largest cohort in europe in COVID period.

The paper is well written. Statistics were performed rigorously

Can you support few precision

1 can you explain why recent lung transplant seems to be less affected and the high proportion of preexisting CLAD. Is a particularity of your global cohort ?

2 in discussion line 290 : "hospital stays were significately shorter". This result may be explain due to the therapeutics engaged in lung transplant patient (dialysis - ECMO) and not in the general population.

3 line 346 you assay that the lack of control group. Can you develop this point. You can realize propensive score with your cohort, especially on CLAD

6. PLOS authors have the option to publish the peer review history of their article (what does this mean?). If published, this will include your full peer review and any attached files.

Reviewer #1: No

Reviewer #2: **Yes: **geoffrey Brioude

---

## [Author Response · Author response to Decision Letter 0]

16 Aug 2021

Rebuttal letter – Response to Reviewers

Reviewer #1:

The subject included in the manuscript is of the utmost importance for broad medical community. Firstly, COVID-19 is the novel disease of unknown long-term consequences. Secondly, undoubtedly it will be present in population for many years, if not decades. Therefore every report sheds a new light on the condition we will be facing in the years to come. It is reviewer satisfaction to be able to congratulate the Authors. The forwarded paper is meticulous, comprehensive, and precise. It is astounding that out of more than 1000 LTx individuals, solely more then 30 have gotten ill. Definitely this indicates accurate measures introduced to prevent them from being infected, some of them you have mentioned in the manuscript. I am quite aware of the leading role of the Hannover Medical School in Europe in lung transplantation. Therefore I got impression the COVID-19 is not affecting seriously this population. It is reassuring message, which perhaps might be more openly emphasized. In the discussion section you have made an extensive comparison with other LTx centers. Generally speaking reading this manuscript has been a teachable moment.

Comment #1:

Thank you very much for your appreciative comments. We are very pleased about this kind feedback.

You suggested giving a little more emphasis on the small amount of lung transplant recipients from our institution affected by COVID-19. In fact, approximately 4 percent of our recipients have been infected by SARS-CoV-2 during the observation period what is comparable to the numbers in the German general population. However, the estimated number of unreported / undetected cases might be considerably higher in the general population so we added the following sentence to our discussion:

Lines 283-284: The incidence of SARS-CoV-2 infections in our patient-cohort during the observation period was similar to nationwide German incidence (figure 3). However, it might be possible that the number of unreported / undetected cases were considerably higher.

Reviewer #2:

Thank you for submititng this retrospective analysis of one of the largest cohort in europe in COVID period. The paper is well written. Statistics were performed rigorously. Can you support few precision:

Comment #2:

We thank you for the valuable comments which helped to improve the manuscript. We have addressed all your comments below and provided explanations where information was requested.

1) can you explain why recent lung transplant seems to be less affected and the high proportion of preexisting CLAD. Is a particularity of your global cohort ?

Thank you for these questions. Chronic lung allograft dysfunction (CLAD) is the most important limiting single factor of long-term survival after lung transplantation, accounting for >40% of deaths beyond the first year.1,2 The amount of lung transplant recipients with pre-existing CLAD in our work seems not to be unexpectedly high as CLAD is a major issue in these patients. Almost one third of our reciepints alive are affected with CLAD. As presented in figure 3, the incidence trend of SARS-CoV-2 infections in our cohort was very similar to that from the German general population.

We added a sentence to the manuscript to highlight the relevance of CLAD in lung transplant recipients:

Lines 302-303: The most important limiting single factor of long-term survival in LTx recipients is CLAD, responsible for >40% of deaths beyond the first year (17).

1) Verleden GM, Glanville AR, Lease ED, Fisher AJ, Calabrese F, Corris PA, et al. Chronic lung allograft dysfunction: Definition, diagnostic criteria, and approaches to treatment-A consensus report from the Pulmonary Council of the ISHLT. J Heart Lung Transplant. 2019 May;38(5):493-503. doi: 10.1016/j.healun.2019.03.009. Epub 2019 Apr 3. PMID: 30962148.

2) Van Herck A, Frick AE, Schaevers V, Vranckx A, Verbeken EK, Vanaudenaerde BM, et al. Azithromycin and early allograft function after lung transplantation: A randomized, controlled trial. J Heart Lung Transplant. 2019 Mar;38(3):252-259. doi: 10.1016/j.healun.2018.12.006. Epub 2018 Dec 14. PMID: 30686699.

2) in discussion line 290 : "hospital stays were significately shorter". This result may be explain due to the therapeutics engaged in lung transplant patient (dialysis - ECMO) and not in the general population.

Thank you for this comment. We added this potential explanation to the discussion section:

Lines 271-274: In this work, in-hospital mortality was 22% (compared to 46% in our study) and hospital stays were significantly shorter (14 vs. 19 days) what might potentially be explained due to the therapeutics engaged in LTx recipients (e.g. exctracorporeal therapies) rather than in the general population.

3) line 346 you assay that the lack of control group. Can you develop this point. You can realize propensive score with your cohort, especially on CLAD

Thanks again for this important consideration and the excellent statistical discussion point. The patient cohort affected by COVID-19 is rather small. We agree that this is a limitation for all forms of multivariate analyses used in small sample sizes. We have recognized propensity score matching as an in-vogue method but we must at the same time acknowledge inherent limitations of this method. Indeed, we discussed to use this method beforehand instead of traditional covariate adjustment. After statistical advice and following recent guidance for statistical reporting we decided against using propensity score matching.3 No change in the revised manuscript regarding this point.

3) Harhay MO, Donaldson GC. Guidance on Statistical Reporting to Help Improve Your Chances of a Favorable Statistical Review. Am J Respir Crit Care Med. 2020;201(9):1035-1038. doi: 10.1164/rccm.202003-0477ED.

---

## [Decision Letter · Decision Letter 1]

13 Sep 2021

COVID-19 in lung transplant recipients - risk prediction and outcomes

PONE-D-21-23993R1

Dear Dr. Kamp,

We’re pleased to inform you that your manuscript has been judged scientifically suitable for publication and will be formally accepted for publication once it meets all outstanding technical requirements.

Kind regards,

Academic Editor

PLOS ONE

Additional Editor Comments (optional):

Reviewers' comments:

Reviewer's Responses to Questions

**Comments to the Author**

1. If the authors have adequately addressed your comments raised in a previous round of review and you feel that this manuscript is now acceptable for publication, you may indicate that here to bypass the “Comments to the Author” section, enter your conflict of interest statement in the “Confidential to Editor” section, and submit your "Accept" recommendation.

Reviewer #1: All comments have been addressed

Reviewer #2: All comments have been addressed

2. Is the manuscript technically sound, and do the data support the conclusions?

Reviewer #1: Yes

Reviewer #2: Yes

3. Has the statistical analysis been performed appropriately and rigorously? 

Reviewer #1: Yes

Reviewer #2: Yes

4. Have the authors made all data underlying the findings in their manuscript fully available?

Reviewer #1: Yes

Reviewer #2: Yes

5. Is the manuscript presented in an intelligible fashion and written in standard English?

Reviewer #1: Yes

Reviewer #2: Yes

6. Review Comments to the Author

Reviewer #1: (No Response)

Reviewer #2: Tak you for this answer.

All the comments were discussed and appropriate modifications done.

No further modifications

7. PLOS authors have the option to publish the peer review history of their article (what does this mean?). If published, this will include your full peer review and any attached files.

Reviewer #1: No

Reviewer #2: **Yes: **Brioude Geoffrey

---

## [Editor Report · Acceptance letter]

15 Sep 2021

PONE-D-21-23993R1 

COVID-19 in lung transplant recipients - risk prediction and outcomes 

Dear Dr. Kamp:

I'm pleased to inform you that your manuscript has been deemed suitable for publication in PLOS ONE. Congratulations! Your manuscript is now with our production department. 

Kind regards, 

on behalf of

Dr. Robert Jeenchen Chen 

Academic Editor

PLOS ONE